# An Empirical Investigation of Environmental Turbulence and Fear in Predicting Entrepreneurial Improvisation

**Sara Shabbir [1], Rizwan Qaiser Danish [2,*], Muqqadas Rehman [2], Muhammad Hasnain [2] and Humaira Asad [2]**

[1]   Hailey College of Commerce, University of the Punjab, Lahore 54590, Pakistan; saraadeel16@gmail.com
[2]   Institute of Business Administration, University of the Punjab, Lahore 54590, Pakistan;
      muqqadasrehman@ibapu.edu.pk (M.R.); hasnain5573019@gmail.com (M.H.); humaira@ibapu.edu.pk (H.A.)
*    Correspondence: rqdanish@ibapu.edu.pk

**Abstract:** This study is designed to test an empirical investigation of the environmental turbulence and fear in predicting entrepreneurial improvisation in small and medium enterprises (SMEs) in Pakistan. This research aims to analyze whether the emotional response of fear drives the decision-makers of SMEs towards the use of improvisation strategy in a turbulent environment in an uncertain business world or not. The target population for this study includes owners, managers, and directors of SMEs listed in the Chambers of Commerce of capital cities of Pakistan. Data were collected through a quantitative survey from 433 respondents by using the cluster sampling technique. Structural equation modeling (SEM) was used to test the hypotheses of the study and conduct path analysis. The findings of this study reveal that environmental turbulence leads the decision-makers towards the use of improvisation strategy by managing their emotions of fear rather than sticking to the plans in SMEs in Pakistan. Moreover, the emotional response of fear in entrepreneurs mediates the relationship between environmental turbulence and entrepreneurial improvisation in SMEs in Pakistan. This study contributes to the field of SMEs by suggesting that entrepreneurs can compete in the frequently changing business world through improvisation. This study helps the Small and Medium Enterprises Development Authority (SMEDA) to understand the importance of entrepreneurial improvisation during uncertainties in the complex business environment, which leads the organization towards innovation.

**Keywords:** environmental turbulence; entrepreneurial improvisation; fear; small and medium enterprises; Pakistan



## 1. Introduction

SMEs play a strategic role in running the national economy [1,2]. They are generally defined as the firms that do not have more than 500 employees. They are considered the backbone of the economy. Their role is crucial for the economic growth of the country. The challenges faced by SMEs have become intense due to the industrial era. The business world faces numerous changes in the environment. Changes such as globalization, information technology, and regulation changes cause turbulence in the environment [2,3]. Environmental turbulence refers to the extent of instability, variability, and predictability that is reflected in the novelties and complex variations in the environment [4,5]. Therefore, organizations must be careful of all those unexpected changes, as they can harm their value if they are overlooked [6]. It means that environmental turbulence can lead to organizational responsiveness, organizational learning, innovation, and competitive advantages [7]. The company must have leadership in technology to obtain success in this surprising environment. In this environment, the growth of advanced technology and new industry takes place like a streak of lightning. Additionally, customers are very keen to pay for the advanced products despite the regular products [8]. Henceforth, companies need to have a full grip on modern technology.

Two important types of environmental turbulence that affect modern business organizations are technology turbulence and market turbulence. A turbulent environment greatly impacts the companies, as there is a lot of pressure on the companies during instability in the environment because of the emergence of innovative technology and new product development in the highly competitive market [9]. The whole staff becomes involved in finding appropriate solutions, due to which the company environment also gets disturbed. As a result, feelings of fear and anxiety are created in such an unstable environment. Because of this fear and anxiety, some workers carry out wrong steps instead of finding the correct solution, making things worse and difficult to solve [10].

Traditionally, entrepreneurs have been supposed to be "causal thinkers". In other words, when launching a business, it is presumed that the future can be predicted, and the entrepreneurs make plans to achieve predefined objectives. This is causal thinking since it is based on prearranged events. Entrepreneurship does not go in the same way, as the future is unpredictable and uncertain. In a complex and ambiguous business world, there is a need to think out of the box. An entrepreneur must adopt an effectual thinking framework rather than causal thinking during change or turbulent conditions [11]. The process of entrepreneurship is full of risks and uncertainties. Original and predefined entrepreneurial plans may not be used in today's intense competition and continuously changing environment. An entrepreneur must be able to decide according to the situation and adapt to every change in order to compete with others [12]. In reality, entrepreneurs have to sacrifice their pre-planning to make their present and future secured in this changing environment where there is high uncertainty, old plans do not work, and they can be disastrous if they are not modified under the present situations [13].

The business world is facing many turbulences in the environment related to the market and technology. To cope with such turbulences, entrepreneurs must intelligently manage their emotions and adopt strategic improvisation to survive. Henceforth, this research helps entrepreneurs to change their thinking framework from static to effective for making decisions in an uncertain business world. Furthermore, it helps entrepreneurs to think differently during turbulent situations by managing their emotions rather than sticking to their plans.

The fundamental aim of this cross-sectional study is to identify the association between environmental turbulence and entrepreneurial improvisation. It is checked through hypothesis testing of direct relationships. This study also aims to determine the impact of fear during environmental turbulence. Furthermore, this study is also designed to determine whether the emotion of fear mediates the relationship between environmental turbulence and entrepreneurial improvisation or not. This mediation analysis is performed by applying bootstrapping strategy. Self-administered online questionnaires are used for conducting this quantitative study.

This model is needed at a time to make a distinction and become successful in the turbulent business environment, as environmental turbulence is inevitable in the dynamic business world. This study contributes in many ways. It gives an insight to the entrepreneurs that they can define a working environment that is favorable for emotional awareness and attentiveness to uncertain situations, especially in technology-based SMEs where dynamic changes occur in the environment. During these situations, entrepreneurs can manage fear and anxiety caused by environmental turbulence and direct those emotions of fear and anxiety towards constructive and positive activities such as improvisation.

Therefore, the research question of this study is: "Do environmental turbulence and emotional responses such as fear predict entrepreneurial improvisation?" Therefore, the research objectives of the study are as follows:

(1) To identify the relationship between environmental turbulence and entrepreneurial improvisation.
(2) To determine whether environmental turbulence leads toward emotional responses such as fear.
(3) To investigate the impact of fear on entrepreneurial improvisation.

(4) To analyze whether emotional responses such as fear mediate the relationship between entrepreneurial improvisation and environmental turbulence or not.

## 2. Literature Review and Hypotheses Development

### 2.1. Environmental Turbulence

Environmental turbulence takes place when there is an existence of unanticipated and unpredictable changes. Some hurdles are uncontrollable, and it becomes impossible for the company to control them, while there can also be some instability that can be turned into opportunities. It means that environmental turbulence can lead to organizational responsiveness, organizational learning, innovation, and competitive advantages. The enterprises working on a larger scale face environmental turbulence more than the smaller enterprises. The greater the resources of a firm, the greater will be the environmental turbulence. Environmental turbulence works as a hurdle that can be raised in the company because of the changes in the environment [7]. Managers are responsible for taking care of these changes so that immediate actions can be taken by utilizing the opportunities and minimizing the chances of threats. Hence, the managers must make those strategies that can deal with internal and external issues and work in changing environments. Environmental turbulence occurs when changes in the environment take place because of the emergence of new technologies, changing economic and political situations, modifications in society's values, and shifts in customers' demands [14].

Technological turbulence and market turbulence are the main types of environmental turbulence. Technological turbulence means the change in the rate of technology in the industry. It is the most influential change in the companies, as this external factor greatly influences the production of the whole company [15]. When some new technology is introduced, or a new product is formed by restructuring the existing technology, then it impacts the other competitors of the same industry. Some firms become able to accept that challenge due to their enriched resources, while some firms face a downturn due to their limited resources and inappropriate strategies [16]. Technological turbulence can be explained as an external environment factor of the firm, and it depicts the modification in the means of production in the company and the development of innovative products [17].

When the stock market goes up and down unpredictably, then it creates market turbulence. It is said that market turbulence takes place when there is a rate of change in the number of buyers and their purchasing preferences [18]. Market turbulence is usually created when there is a changing trend in the climate of the economy, changes in needs and demands of buyers, and continuous advancement in technology. The two major causes of market turbulence are the driving forces: one is the consumer, and the second is the new technology [19].

The turbulent environment is the state of the environment that is complex due to the presence of an international market; the entrance of outside competitors; the emergence of new variations in technology, rapidly changing needs, demands, and attitudes of customers; and changing governmental, economic, and societal trends [20]. Most companies operate in a discontinuous environment by facing challenges and complexities. The frequency of change is also greater in this environment, due to which many companies are unable to respond to those particular changes, and then they face losses [7]. The company must have leadership in technology to obtain success in this surprising environment. In this environment, the growth of advanced technology and new industry takes place like a streak of lightning, and customers are also willing to pay for the advanced products despite the regular ones [8].

It is argued that enterprises operating under a turbulent environment have a higher ability to improvise their behaviors and approaches to respond to unplanned happenings and fluctuations of the environment. Therefore, environmental turbulence seems to be an essential predictor of entrepreneurial improvisation. A study was conducted in China to examine the antecedents of entrepreneurial improvisation in which the regulatory focus was regarded as a significant predictor of entrepreneurial improvisation [21]. Environ-

mental turbulence has the potential to determine the extent to which regulatory focus affects entrepreneurial improvisation [21]. It means that the regulatory focus tends to determine entrepreneurial improvisation in the presence of environmental turbulence gradually. This is because the environmental turbulence enables the entrepreneur to respond to uncertain and prompt changes, so that entrepreneurial improvisation enhances. The significant role of environmental turbulence was discussed by [22,23] in determining entrepreneurial improvisation by highlighting the potential of environmental turbulence to affect the proactivity, responsiveness, and pace of decision-making of entrepreneurs.

The environmental uncertainties and rapid fluctuations may make the entrepreneurs and management of the firm stressed and fearful. The firm may become worried about its competition, survival, and growth in the market. Several studies are found in the literature that supports the viewpoint that environmental turbulence can give rise to different emotions, e.g., stress and fear [24,25]. Fear imparts a harmful impact on the creativity of the firm. When there is time pressure, the stress level will be higher, and the creativity level will be lower. Ultimately, these emotional responses affect the decision-making process in the firms. Due to stress and fear, the capabilities of team members are also affected, such as information-processing and handling situations. SMEs face these emotions when there is a high workload pressure due to time constraints and challenging activities. These emotional responses play important roles in determining the ultimate influence of environmental turbulence on entrepreneurial improvisation because these emotions push the entrepreneur to think and react quickly to the new changes and market trends. However, if the entrepreneur does not think innovatively and dynamically, there will be a question mark on the success of the firms. The following hypotheses can be drawn based on the above-mentioned theoretical premise:

**Hypothesis 1 (H1).** *Environmental turbulence has a significant positive impact on entrepreneurial improvisation.*

**Hypothesis 2 (H2).** *Environmental turbulence and fear are positively related.*

*2.2. Entrepreneurial Improvisation*

Today, entrepreneurial improvisation is an important and novel strategy to deal with a dynamic environment and uncertain external events effectively. It is a relatively new concept and is growing rapidly in many small and medium enterprises. Several past studies, e.g., [9,26,27], regarded entrepreneurial improvisation as an integral approach that needed to be adopted by entrepreneurs to cope with uncertain events spontaneously. Improvisation does not guarantee the enterprises' success or failure. However, the literature supports that turbulence in the business environment leads to innovation if the entrepreneurs adopt an improvisation strategy in SMEs. If the resources of a firm are stronger, then the tendency toward entrepreneurial improvisation will be higher compared to the smaller firms. Various internal and external factors impact entrepreneurial improvisation in SMEs. The process of entrepreneurship is full of risks and uncertainties. Original and predetermined entrepreneurial plans may not be applied in today's highly competitive and continuously diversifying environment. The entrepreneur must be able to decide according to the situation and adapt to every change to compete with others [12]. In reality, entrepreneurs have to sacrifice their pre-planning to make their present and future secured, as in this changing environment where there is high uncertainty, old plans do not work and can be disastrous if they are not modified under the present situations [13]. This uncertainty creates an alarming state for entrepreneurs, as they have to make themselves ready for all situations and circumstances that can happen at any time. Some entrepreneurs make their initial strategies in such a way that they can be applied at any difficult time or situation.

In addition, those strategies are usually innovative and can solve every problem [28]. Environmental turbulence as a predictor of entrepreneurial improvisation has been taken under current consideration. Prior studies emphasized that the external factors and envi-

ronment play a substantial role in driving the creativity and innovation of an enterprise that are the key components of entrepreneurial improvisation. For example, it is suggested that the factors of creativity and impulsiveness are needed by the firm to efficiently and quickly respond to the external market shift [29]. This means that entrepreneurial improvisation is significantly linked with external factors and the environment because external factors and the turbulent environment promote the impulsiveness and creativity that are the key components of entrepreneurial improvisation.

Some firms are resilient in facing challenges. They try to behave proactively and oppose adversity. Resilience is a strategy to deal with risk and uncertainty. The firms that have the potential to bounce back the uncertainty and unpredictability are considered resilient firms. Resilience leads to organizational success. Additionally, entrepreneurial improvisation is associated with resilience. There are some crises faced by the enterprises. Crises threaten the values of enterprises that are of high value. They are the process that is extended by time. They are mostly unpredictable and unexpected. They give a constrained time for making a response. Therefore, it is essential for firms to adopt effective crisis management techniques. The firms try to bring the situations back to normal. Firms can adopt communication and coordination to respond to environmental changes and crises. Effective leadership styles are helpful in responding to changes. Leaders know how to combat dynamic changes and crises in the firms.

Strategic managers need to improvise since, without doing so, they cannot develop their organizations. The firms that achieve a high level of improvisation are considered to be rational. The managers who think in a rational way always prefer improvisation. In emerging economies, improvisation is the need of the hour. Entrepreneurial improvisation is considered an effective strategy for corporate executives. Efforts are made for entrepreneurial improvisation if the firm has good corporate governance. However, if entrepreneurial improvisation does not positively affect communication, teamwork, decision making, social interaction, and dealing with fear and anxiety, then it is considered irrational. If it is done in an irrational way, it is not favorable for the firm. Additionally, corporate governance would like to mitigate such irrational entrepreneurial improvisation.

### 2.3. Fear

Businesspeople are humans and are not simply "employed hands". They carry their heads and hearts with particular qualities, interests, estimations, and capacities [30]. Although environmental turbulence tends to push entrepreneurs to adopt innovative approaches and behaviors to respond to unplanned and unpredictable changes effectively, it also gives rise to various emotions. A study was conducted in Turkey to investigate the role of stress during a turbulent environment in the relationship between intuition and decision-making [24]. The findings of that study suggested that a low level of stress leads to better decision-making during a turbulent environment. On the whole, negative feelings (for example, fear) have a high adaptive value, as they empower people to limit thought–activity collections in undermining circumstances and proficiently develop answers to the issues [31]. In the light of the available literature, examination suggested that individuals who are high in quality constructive influence will react to fear because of emergencies in a progressively valuable way to such an extent that these individuals will deal with their all feelings more adequately and thus will improvise more.

Turbulence in the business environment disturbs the enterprises because high pressure is placed on individuals during uncertain conditions, as there are sudden technological breakthroughs and new product developments in the competitive business world [9]. Here, firm size also matters. If the firm is working at a broader level, the environment will be more turbulent. Due to a highly turbulent environment, negative feelings such as fear and anxiety will have a comparatively greater impact. When some new product is introduced, then each company starts trying to make something more unique than others. In this way, the period of turbulence becomes more enhanced. This condition creates fear among the companies, as in this way, they have to face a lot of cost and time to regain their position

and make their position strong among competitors [32]. The panic situation can also be raised if any of the competitors provide a unique product during the turbulent period and the other company has limited resources to compete [10,33].

Among different emotional responses, fear during a turbulent period is an essential emotional response that plays an integral role in the relationship between environmental turbulence and entrepreneurial improvisation. It means that emotional responses to the turbulent environment such as fear, stress, and anxiety can enhance or reduce entrepreneurs' effectiveness to improvise [34]. However, the role of emotional responses in a turbulent environment and their impacts on entrepreneurial improvisation were largely ignored in the existing literature. Uncertain business conditions or turbulent environments create fear in entrepreneurs. Based on this premise, the following hypotheses are postulated:

**Hypothesis 3 (H3).** *Fear is positively associated with entrepreneurial improvisation.*

**Hypothesis 4 (H4).** *Fear during environmental turbulence significantly mediates the relationship between environmental turbulence and entrepreneurial improvisation.*

*2.4. Conceptual Framework*

The hypothesized model is presented in Figure 1:

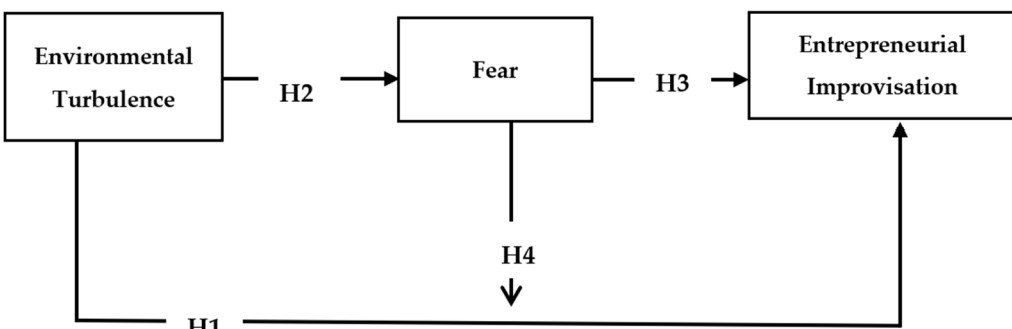

**Figure 1.** A conceptual framework describing the impact of environmental turbulence on entrepreneurial improvisation through fear.

**3. Research Methodology**

*3.1. Participants and Procedure*

The data were collected from the owners, managers, and directors of SMEs listed in the Chambers of Commerce of capital cities of Pakistan. Their online existence was assessed through reliable websites. SMEs contribute 40% of the gross domestic product in less-developed countries such as Pakistan, and they consist of 90% of the business organizations [35,36]. The cluster sampling (non-probability sampling) technique was used to draw the sample size for the study. A total of 590 respondents were randomly selected from clusters in the capital cities of Pakistan registered with the Chambers of Commerce. The data were collected at a single point in time due to time constraints; hence, this was a cross-sectional study. The approach of this study was quantitative since all the variables were well-developed and well-measured. The unit of analysis for this research was the managers, owners, and directors of SMEs, as they have primary responsibility for making decisions related to the adaption of improvisation strategy when there are uncertainties in the business world. Since different types of activities can affect the responses, the responses were gathered during regular working hours. Hence, the biases in responses were not there. SMEs were operating in consumer staples, communication services, information technology, the consumer discretionary sector, and finance. The quantitative survey was used to collect responses from the owners, managers, and directors of SMEs of Pakistan by using a 5-point Likert scale anchored by 1 (strongly disagree) and 5 (strongly agree). The data collection process took almost two months. A total of 433 responses were received back,

which shows a 73.39% usable response rate. This study was conducted with minimal researcher interference in the work settings, which is called a non-contrived study [37].

### 3.2. Measures

Data collection methods include interviews, observations, surveys, and questionnaires depending on the type of research question and the study type [37]. This research is primary in nature since the data were gathered from primary sources through online questionnaires. To conduct this research, data were collected by distributing online questionnaires, which were developed by using a Likert scale containing 26 items. The scales for the variables of research were adopted from well-established studies, which are mentioned below.

#### 3.2.1. Environmental Turbulence

Environmental turbulence means the extent of instability, variability, and predictability that is revealed in the novelties and complex variations in the environment [4,5]. Environmental turbulence was measured using a 9-item scale given by [21]. A sample item is "a large number of products in our market have been made possible through technological breakthroughs".

#### 3.2.2. Entrepreneurial Improvisation

An entrepreneurial improvisation is a crucial approach that needs to be adopted by entrepreneurs to cope with uncertain events [9,26,27]. Entrepreneurial improvisation was measured using a 12-item scale suggested by [38]. A sample item is "I take risks in terms of producing new ideas in completing projects".

#### 3.2.3. Fear

The unplanned and unpredictable changes give rise to various emotions, and fear is one of them. Fear was measured using a 5-item scale applied by [39]. A sample item is "I feel anxious when I hear about impending changes".

#### 3.2.4. Control Variables

Demographics of the instrument include the gender of the respondents by using the nominal scale, the actual age of the respondents, marital status of the respondents, and the nature of enterprise of SME, whether SME is a manufacturing, trading, or services concern.

### 3.3. Data Analysis Strategy

Statistical Package for the Social Sciences (SPSS) version 22 and Analysis of Moment Structure (AMOS) version 26 were used for the analysis of data collected through the online questionnaires. SPSS was used for the preliminary analysis, and AMOS was used for conducting SEM and path analysis.

## 4. Results and Analysis

### 4.1. Preliminary Analysis

The preliminary analysis includes a data normality test. The preliminary analysis also consists of the statistics and distribution of data such as mean, standard deviation, and the interpretation of demographics used in the research to analyze the frequency of demographical aspects in the study.

### 4.2. Demographical Analysis

Demographical information is extracted from the filled online questionnaires, which were received back using SPSS. Results are given as under in Table 1:

**Table 1.** Demographic profile of the respondents.

| Characteristics | Classification | Frequency | Percentage (%) |
|---|---|---|---|
| Gender | Male | 174 | 40.2 |
| | Female | 259 | 59.8 |
| | Total | 433 | 100 |
| Marital status | Married | 221 | 51 |
| | Single | 212 | 49 |
| | Total | 433 | 100 |
| Age (Years) | 20–30 | 202 | 46.66 |
| | 30–40 | 88 | 20.32 |
| | 40–50 | 87 | 20.09 |
| | Above 50 | 56 | 12.93 |
| | Total | 433 | 100 |
| Nature of Enterprise | Manufacturing | 129 | 29.8 |
| | Trading | 146 | 33.7 |
| | Services | 158 | 36.5 |
| | Total | 433 | 100 |

Frequencies and percentages of gender, marital status, age, and nature of enterprise are calculated by using SPSS. Data are collected from 433 respondents. The majority of the respondents were female, which is 59.8%. The majority of the respondents were married, which is 51%. Most of the respondents were falling in the age bracket of 202, which is 46.66%. The majority of the respondents were from services-providing enterprises, which is 36.5%.

*4.3. Descriptive Statistics and Correlation Analysis*

The descriptive statistics, normality tests, and correlation analysis performed in SPSS are presented in Table 2:

**Table 2.** Descriptive statistics, normality tests, and correlation analysis.

| Variables | Mean | Standard Deviation | Skewness | Kurtosis | Correlation | | |
|---|---|---|---|---|---|---|---|
| | | | | | 1 | 2 | 3 |
| 1.Environmental Turbulence | 2.9841 | 0.32381 | 0.283 | 0.531 | 1 | | |
| 2.Fear | 3.1095 | 0.65414 | −0.341 | −0.924 | 0.226 ** | 1 | |
| 3.Entrepreneurial Improvisation | 3.0701 | 0.43059 | 0.037 | −0.491 | 0.455 ** | 0.610 ** | 1 |

** Correlation is significant at the 0.01 level (2-tailed).

Table 2 depicts that the mean values range from 2.9841 to 3.1095. It shows that the majority of the respondents were either neutral or agreed in their responses. All the standard deviation values are less than 1, and it shows that the model is free from dispersion. The data normality was checked through skewness and kurtosis. The collected data show a normal distribution. The skewness value of the data must be within +1 and −1 [40]. Kurtosis estimates must lie between +3 and −3 for normal distribution [41]. As shown in Table 2, the skewness and kurtosis values fall within the prescribed limits. Hence, the data for this study is normally distributed and fit for further analysis.

Correlation analysis shows that a significant and positive relationship exists among all the constructs of the study. Environmental turbulence is positively associated with fear ($r = 0.226$, $p < 0.01$) and with entrepreneurial improvisation ($r = 0.455$, $p < 0.01$). Fear is positively related to entrepreneurial improvisation ($r = 0.610$, $p < 0.01$). Since all the correlation values are less than +0.85, there is no multicollinearity issue in the data [42].

*4.4. Structural Equation Modeling*

SEM is a well-recognized analysis technique in social sciences and management sciences, as it simultaneously gives a structural model and measurement model [43]. SEM is

a way to analyze and calculate the relationships by illustrating the available data to prove hypotheses. It is a combination of factor analysis and multiple regression analysis, as it includes regression, factor evaluation, and path evaluation. For this study, SEM analysis was performed to test the model by using AMOS version 26. Confirmatory factor analysis (CFA) was performed to prove the convergent validity and composite reliability of the data. Discriminant validity is established by taking root mean square of AVE and bivariate correlation. The overall fitness of the model is tested by calculating CMIN/D. F., the goodness of fit index (GFI), average goodness of fit index (AGFI), comparative fit index (CFI), and root mean square error of approximation (RMSEA) values.

### 4.5. Confirmatory Factor Analysis

CFA is an evaluating model and a form of structural equation model. In this study, CFA was conducted with the help of AMOS version 26 to analyze individual factors. Some items were deleted to establish CFA whose factor loading value was less than 0.30 [44]. Table 3 is developed to ensure the validity and reliability of the data. It demonstrates the factor loading values of all items. It can be observed in Table 3 that the factor loadings of all the items are greater than 0.30, which means that all the items are loaded well on the respective constructs. The composite reliability (CR) threshold value must be greater than 0.70 to establish the convergent reliability [45]. CR values for all the constructs of the study are greater than 0.70, which means that the convergent validity of the constructs is established. Cronbach's Alpha values of all the constructs mentioned in Table 3 are greater than 0.70, which means all the constructs of the measuring instrument are reliable enough to measure the construct under consideration.

**Table 3.** Factor loading values ($\lambda$) of CFA, composite reliability, and Cronbach's Alpha.

| Constructs | Items | Factor Loadings | CR | Cronbach's Alpha |
|---|---|---|---|---|
| Environmental Turbulence (ET) | ET1 | 0.388 | 0.802 | 0.763 |
| | ET2 | 0.694 | | |
| | ET3 | 0.560 | | |
| | ET4 | 0.769 | | |
| | ET5 | 0.610 | | |
| | ET6 | 0.603 | | |
| | ET7 | 0.438 | | |
| | ET8 | 0.539 | | |
| Fear (F) | F1 | 0.967 | 0.808 | 0.744 |
| | F2 | 0.314 | | |
| | F3 | 0.996 | | |
| Entrepreneurial Improvisation (EI) | EI1 | 0.964 | 0.748 | 0.764 |
| | EI2 | 0.367 | | |
| | EI3 | 0.644 | | |
| | EI4 | 0.751 | | |
| | EI5 | 0.307 | | |
| | EI6 | 0.314 | | |

### 4.6. Discriminant Validity

Discriminant validity was analyzed by comparing the bivariate correlation among constructs and the square root of average variance extracted (AVE) of each construct [43]. "AVE is a more conservative measure than CR; without AVE, a researcher cannot conclude that the discriminant validity is established" [46]. All the diagonal values in Table 4 exceed the other values in the same row; therefore, it can be assumed that discriminant validity was held in the data [47]. Bivariate correlation values of all the constructs are significant, which means that a significant positive correlation exists among the variables of the study. Hence, the discriminant validity was confirmed.

**Table 4.** Discriminant validity.

| Variables | Environmental Turbulence (ET) | Entrepreneurial Improvisation (EI) | Fear (F) |
|---|---|---|---|
| Environmental Turbulence (ET) | 0.586 [a] | | |
| Entrepreneurial Improvisation (EI) | 0.455 [b] | 0.763 | |
| Fear (F) | 0.226 | 0.610 | 0.805 |

Note: [a] AVE square root in diagonal. [b] Bivariate correlation among constructs in off-diagonal.

### 4.7. Model Fitness Summary

Table 5 shows the overall model fitness summary for the study. The value of CMIN/D. F. is 3.650, which should be less than or equal to 5 for acceptance. The GFI value of the model is 0.901 (should be greater than or equal to 0.90), and the AGFI value is 0.861 (should be greater than or equal to 0.80) [48–50]. Both values for the model are greater than their respective thresholds, which indicates the goodness of fit of the model [47]. The CFI value of the model is 0.917, which should be greater than or equal to 0.90. It also depicts the goodness of fit of the model. The value of RMSEA should be less than or equal to 0.08 for the fitness of the model, and in this study, the value of RMSEA is 0.078, which is also reliable and beneficial for model fitness [47]. The model fitness summary reveals that the model is perfectly fit for further analysis, i.e., to test the hypotheses of the study and to draw inferences for the research conducted.

**Table 5.** Model fitness summary.

| Model Fit Indices | Structural Model | Thresholds |
|---|---|---|
| CMIN/D. F. | 3.650 | <5 |
| GFI | 0.901 | ≥0.90 |
| AGFI | 0.861 | ≥0.80 |
| CFI | 0.917 | ≥0.90 |
| RMSEA | 0.078 | <0.08 |

The structural model is presented in Figure 2:

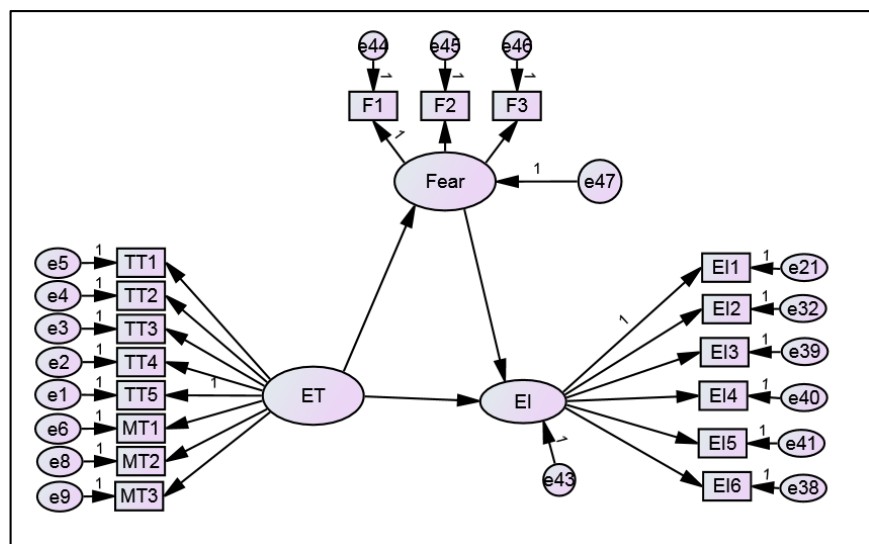

**Figure 2.** Structural model.

### 4.8. Hypotheses Testing—Direct Effects

As all the model parameters are fully satisfied, the structural model is developed to assess the relationships between variables in AMOS in Figure 2. Table 6 shows the

direct relationships among the variables of the study. Environmental turbulence has a positive and significant impact on entrepreneurial improvisation ($\beta$ = 0.952, $p$ < 0.001). Environmental turbulence has a positive and significant impact on fear ($\beta$ = 0.749, $p$ < 0.001). Fear has a positive impact on entrepreneurial improvisation ($\beta$ = 0.571, $p$ < 0.001). Therefore, H1, H2, and H3 of the study are accepted.

**Table 6.** Hypotheses testing—direct effects.

| Hypotheses | Estimates | *p*-Value |
|---|---|---|
| Environmental Turbulence → Entrepreneurial Improvisation (H1) | 0.952 | *** |
| Environmental Turbulence → (Fear (H2) | 0.749 | *** |
| Fear ( Entrepreneurial Improvisation → (H3) | 0.571 | *** |

Note: *** *p*-value < 0.001.

### 4.9. Mediation Analysis

AMOS is used to perform mediation analysis in structural equation modeling. A re-sampling strategy, bootstrapping, was applied for mediation analysis [51]. Table 7 explains the summary of the mediation analysis. Mediation analysis was performed to find out the impact of mediating variable (fear) on the relationship between the independent variable (environmental turbulence) and the dependent variable (entrepreneurial improvisation). Partial mediation of fear is observed in the relationship between environmental turbulence and entrepreneurial improvisation because beta without mediation is positive and significant ($\beta$ = 0.952, $p$ < 0.001), and indirect beta is also positive and significant in the presence of mediating variable of fear ($\beta$ = 0.435, $p$ < 0.001). Hence, H4 is accepted, and it can be interpreted that fear mediates the association between environmental turbulence and entrepreneurial improvisation. It is evident from the results that entrepreneurs become fearful during turbulence in the business environment, but they improvise rather than adopting withdrawal behavior.

**Table 7.** Mediation analysis.

| Variables | Direct Beta without Mediation | Direct Beta with Mediation | Indirect Beta | Mediation Type Observed |
|---|---|---|---|---|
| ET → F → EI (H4) | 0.952 *** | 0.557 *** | 0.435 *** | Partial mediation |

Note: ET, Environmental turbulence; F, Fear; EI, Entrepreneurial improvisation; *** *p*-value < 0.001.

### 4.10. Endogeneity Problem

The endogeneity problem creates a problem in understanding the association among the variables. It arises when the independent variables are related to the error term. This is-sue is common in many fields. There are several methods to deal with the endogeneity problem. First, the independent variable, environmental turbulence was lagged to resolve this issue. It can solve the simultaneity problem. Second, the researcher changed the signs of variables in order to assess the endogeneity issue in regression. It was found that the relationship among all the variables of the study was positive. A positive, strong, and signif-icant correlation was found among all the variables of the study. Moreover, the time-variant control variables were added to examine the relationship among the variables. The control variables are the third factor that simultaneously impacts independent and dependent variables. Finally, the variables of the study were found to have a positive relationship. In short, the endogeneity problem can be mitigated by using any effective strategy [52].

## 5. Discussion: Environmental Turbulence, Entrepreneurial Improvisation, and Fear

The fundamental aim of the study is to identify the association between environmental turbulence and entrepreneurial improvisation and analyze the mediating role of fear in this relationship. It was found that environmental turbulence has a positive impact on entrepreneurial improvisation. Moreover, it was proven that environmental turbulence positively impacts the emotion of fear, and fear positively impacts entrepreneurial improvi-

sation. Furthermore, it was observed that fear partially mediates the relationship between environmental turbulence and entrepreneurial improvisation. Henceforth, all four hypotheses of the study were fully supported. These findings are consistent with the previous studies. Environmental turbulence has the potential to determine the extent to which regulatory focus affects entrepreneurial improvisation [21]. The regulatory focus determines entrepreneurial improvisation in the presence of environmental turbulence progressively. The reason is that the environmental turbulence makes the entrepreneur capable of responding to uncertain and prompt changes. As a result, entrepreneurial improvisation increases. Environmental turbulence has a significant contribution to determining entrepreneurial improvisation by highlighting the potential of environmental turbulence to impact the proactivity, responsiveness, and pace of decision-making of entrepreneurs [22,23]. Several studies showed the positive influence of environmental turbulence on entrepreneurial improvisation [21,26,53,54]. These researchers discussed the positive influence of different dimensions of environmental turbulence, including market turbulence and technological turbulence, on the proactivity and improvisation of entrepreneurs.

The business environment faces uncertainties and rapid fluctuations, due to which the entrepreneurs and management of the business organizations become stressed and fearful. Due to such an intense environment, business organizations become worried regarding their survival, competitiveness, and market growth. Therefore, the environmental turbulence is the cause of increasing the different emotions such as stress and fear [24,25]. It is highlighted and harmonized by many past scholars that organizations need to evaluate the external environment and embrace proper responses to fluctuations within the external environment in order to endure a competitive advantage and achieve desired outcomes as well as a targeted success [55,56].

Fear is one of the most powerful and significant emotional responses among all the emotions. Fear contributes positively and significantly to the association between environmental turbulence and entrepreneurial improvisation. Hence, an emotional response such as fear can increase or decrease the efficiency of entrepreneurial improvisation [34]. Previous studies have focused on the point that the external environmental factors significantly impact the key components of entrepreneurial improvisation, such as creativity and innovation. The enterprise can efficiently respond to such external environmental factors with the help of impulsiveness and creativity [29]. Therefore, entrepreneurial improvisation has a relationship with these external factors. The reason is that such external factors promote the critical components of entrepreneurial improvisation. Hence, the findings of this study replicate the previous results.

### 5.1. Theoretical and Practical Implications

This study focuses on the cognitive aspect of improvisation. This research theoretically contributes to the existing literature on environmental turbulence, given that uncertain and unpredictable changes in the business environment can make the entrepreneurs experience emotional responses such as fear, and by overcoming that emotion, a business individual may adopt improvisation strategy rather than withdrawal behavior to remain in the market. Many types of research focus on the way that improvisation proves to be beneficial during uncertain situations. For example, improvisation serves as a strategy to deal with problems in the business world when there are limited resources to deal with [57]. When there is time pressure on the business individuals, they use improvisation that leads them to new product development [5]. However, very limited studies are focused on the emotional aspect of environmental turbulence that drives entrepreneurs towards improvisation. This study offers a look into the scenarios that trigger emotions and will lead the entrepreneurs towards improvisation.

The current study contributes practically in the way that when managers and entrepreneurs become fearful during the change in the business world, they may adopt improvisation rather than stick to plans. Entrepreneurs may define such a working environment as favorable for emotional awareness and vigilance to uncertain situations,

especially in technology-based SMEs where the environment dynamically changes. During such conditions, entrepreneurs can cope with fear and anxiety caused by turbulence and direct those emotions of fear and anxiety towards constructive and positive activities such as improvisation. Through improvisation, entrepreneurs may compete in the frequently changing business world. The consequences of this study will also help academic circles and SMEDA to understand the importance of entrepreneurial improvisation during uncertainties in the business environment, which leads the organization towards innovation. In addition, the policymakers should make sure that the procedures related to improvisation, technology, and research should not be unnecessarily high. Therefore, there is a need to develop some new regulations in SMEs.

### 5.2. Limitations and Future Directions

Although the present study contributes to the existing literature by investigating the relationship between environmental turbulence and entrepreneurial improvisation through the mediating role of fear, it still has some limitations. This study examined the small and medium enterprises of the capital cities of Pakistan only. Future research can be conducted on large-scale industries to obtain more generalized results. Future researchers can test and analyze other antecedents and outcomes of improvisation in SMEs and large-scale organizations. The findings of this study are limited to Pakistan only. Future studies can explore the antecedents of SMEs in Western countries. Furthermore, the data were collected for the independent, dependent, and mediating variables at a single point in time (cross-sectional study). Therefore, the determination of causal relationships is difficult. Future studies can collect data for all the variables at multiple points in time (longitudinal study) to investigate the causal relationships.

Linking to the appraisal theory, negative emotions such as fear that arise from uncertain changes in the environment are perceived as threats [58]. Future studies may be conducted by taking other cognition-based responses such as anxiety, upset, distress, or tension [34]. Future researchers may consider other antecedents of improvisation, such as organizational change and crisis. In the future, researchers may use improvisation strategies to make organizations innovative and competitive. Outcomes of improvisation may be considered for future research, as improvisation is just like a skill or tool and does not always guarantee success [59].

### 6. Conclusions

The findings reveal that there are positive as well as significant relationships that exist among entrepreneurial improvisation, environmental turbulence, and fear; additionally, fear mediates this relationship. Henceforth, all the hypotheses of the study are fully supported. These findings suggest that the environment changes rapidly worldwide, and enterprises must have knowledge and strategies to cope with those unexpected events. The results of this study also provide an insight to the entrepreneurs that they must know how to thrive effectively in the dynamic business world by adopting a more flexible and comprehensive approach because this world is a place for those who are adaptive to change rather than stagnant. The phrase "survival of the fittest" is also worth mentioning here, as those enterprises that adapt according to change in the turbulent business environment flourish and sustain [60]. This study addresses the antecedent of improvisation by managing fear to remain in the business game during a turbulent environment. This theoretical model may be expanded and empirically tested in the future by incorporating and identifying more antecedents and consequences of improvisation. Generalizability can be enhanced by observing those antecedents in large-scale firms.

**Author Contributions:** Conceptualization, S.S.; Formal analysis, M.H.; Investigation, H.A.; Methodology, H.A. and M.H.; Project Administration, R.Q.D.; Resources, R.Q.D.; Software, R.Q.D. and M.H.; Supervision, M.R.; Validation, S.S. and M.R.; Visualization, S.S.; Writing—Original Draft, S.S., R.Q.D. and M.R.; Writing—Review & Editing, M.H. All authors have read and agreed to the published version of the manuscript.

**Funding:** This research received no external funding.

**Institutional Review Board Statement:** This study was conducted according to the guidelines of the declaration of Pakistan and approved by the Institutional Review Board of the University of the Punjab on 20 April 2021.

**Informed Consent Statement:** Informed consent was obtained from all subjects involved in the study.

**Data Availability Statement:** The data presented in the study are available on request from the corresponding author.

**Acknowledgments:** The authors would like to acknowledge all the owners, managers, and directors of SMEs who participated in this study. The authors are grateful to the faculty members of the University of the Punjab for their support and guidance.

**Conflicts of Interest:** The authors declare no conflict of interest.

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
