# Peer review of "An Empirical Investigation of Environmental Turbulence and Fear in Predicting Entrepreneurial Improvisation"

_2199-8531, doi:10.3390/joitmc7020157_

Round 1
Reviewer 1 Report
Remove the name of the software from the Abstract. What does it mean that you make a future research suggestion in the abstract? The abstract should give the reader an idea of what you are doing, not what should be done in the future.
Do the types of activities that each SME does, affect the way they responded to the questions? What were your questions? How long did they have to respond? How long was the survey?
You have to define the items that you asked to collect information about your features.
I haven’t seen the normality test results in the paper; you just cited other documents that the values should follow a specific distribution.
Author Response
Thank you for your valuable suggestions!
We have removed the software name from the abstract. We have revised the abstract by removing the suggestions. We have explained different types of activities' effects on responses. Sample questions have been added in measures and time of response has also been mentioned. All the items have been defined. The normality tests have been added along with the values.
Moreover, moderate English changes have been made. The introduction and methods have been improved. The research design, results, and conclusion have been made better as well.
Reviewer 2 Report
The article is well structured and it is consisted of all important elements of scientific article. The topic is interesting for the readers.
Author Response
Thank you for your outstanding remarks related to our manuscript!
You have not suggested us any changes. We have still tried to make our manuscript more worthy. We once again thank you for your appreciation!
With warm regards.
Reviewer 3 Report
attached

Author Response
Thank you for your precious suggestions!
We have improved the English editing of our manuscript. The introduction has been revised. The research design, methods, results, and conclusions have been improved as per your kind suggestions. In the introduction, a preview of research methods and results have been given. The contributions have been added in the introduction clearly. The potential issue of endogeneity has been explained along with the remedies. The effect of firm size has been elaborated. The endogeneity problem has been discussed as per the suggested paper by you. The requirements for the rational improvisation strategy have been mentioned. Simple and short sentences have been used. The conclusion section has been improved.
Reviewer 4 Report
The topic of the paper is pertinent, current and its results can present solutions for this area of knowledge and small and medium enterprises.
The abstract is well prepared and has all the essential contents of the investigation.
The Introduction provides an excellent framework for the research problem, limits the area of the study object and the objectives are consistent with the keywords of the research topic. It would have been interesting for the authors to ask an investigative starting question at the end of the Introduction.
The literature review is sufficient and appropriate to respond to the objectives of the investigation. However, in subsection 2.1 the literature review on the variables of the emotions under study could have been more detailed, namely their consequences. In subsection 2.2 it would have been interesting for the authors to have reviewed other keywords such as crisis management and organizational and behavioral resilience. The hypotheses are well formulated and respond to the objectives of the investigation.
Subsection 3.2 is well prepared, however the authors should mention whether the data collection instrument belongs to the authors or if it comes from other authors.
In section 4, the statistical methods used are suitable for processing the data and responding to the formulated hypotheses. The results of section 4 are presented in an appropriate way and their analysis is correct.
In section 5, the authors discuss the results correctly with other authors. The implications for the theory and practice suggested by the authors are appropriate.
The conclusions are relevant and highlight the main results.
In general, the paper is well structured; there is a coherence between the theme - research objectives - and the scientific methodology followed; the conclusions identify the most relevant results achieved; the references are current.
Author Response
Thank you so much for your appreciation and kind suggestions!
We have added a research question at the end of the introduction. In section 2.1, the literature on emotional responses has been explained further. In section 2.2, the variables such as crisis management and resilience have been introduced. Subsection 3.2 has been improved.
We once again thank you for your valuable review related to our manuscript!
With warm regards.
Round 2
Reviewer 1 Report
I think you are good to go with this paper. Thanks.
Reviewer 3 Report
Well done.